# Learning Robot Soccer from Egocentric Vision with Deep Reinforcement Learning

**Dhruva Tirumala**[1,2]*    **Markus Wulfmeier**[1]    **Ben Moran**[1]    **Sandy Huang**[1]

**Jan Humplik**[1]    **Guy Lever**[1]    **Tuomas Haarnoja**[1]    **Leonard Hasenclever**[1]

**Arunkumar Byravan**[1]    **Nathan Batchelor**[1]    **Neil Sreendra**[3]    **Kushal Patel**[3]

**Marlon Gwira**[3]    **Francesco Nori**[1]    **Martin Riedmiller**[1]    **Nicolas Heess**[1,2]

**Abstract:**

We apply multi-agent deep reinforcement learning (RL) to train end-to-end robot soccer policies with fully onboard computation and sensing via egocentric RGB vision. This setting reflects many challenges of real-world robotics, including active perception, agile full-body control, and long-horizon planning in a dynamic, partially-observable, multi-agent domain. We rely on large-scale, simulation-based data generation to obtain complex behaviors from egocentric vision which can be successfully transferred to physical robots using low-cost sensors. To achieve adequate visual realism, our simulation combines rigid-body physics with learned, realistic rendering via multiple Neural Radiance Fields (NeRFs). We combine teacher-based multi-agent RL and cross-experiment data reuse to enable the discovery of sophisticated soccer strategies. We analyze active-perception behaviors including object tracking and ball seeking that emerge when simply optimizing perception-agnostic soccer play. The agents display equivalent levels of performance and agility as policies with access to privileged, ground-truth state[3]. To our knowledge, this paper constitutes a first demonstration of end-to-end training for multi-agent robot soccer, mapping raw pixel observations to joint-level actions, that can be deployed in the real world.

**Keywords:** robotics, reinforcement learning

## 1 Introduction

In recent years, access to accurate simulators combined with increased computational power has enabled reinforcement learning (RL) to achieve considerable success across both single- and multi-agent settings in robotics, including quadruped parkour [1, 2, 3], robot soccer [4], full-size humanoid locomotion [5], dexterous manipulation [6], and quadrotor racing [7]. While these results are promising, certain simplifying assumptions limit their general applicability. Notably, many of these examples rely on depth sensing [1, 2], which can be expensive; external state estimation [4, 6], which may not always be possible; or specifically-designed modular architectures [3, 7], which often involve domain-specific assumptions; or do not address the problem of exteroception [5].

---

*1. Google DeepMind; 2. University College London (UCL); 3. Proactive Global.

†Correspondence to dhruvat[at]google.com

³Videos of the game-play and analyses can be seen on our website https://sites.google.com/view/vision-soccer.

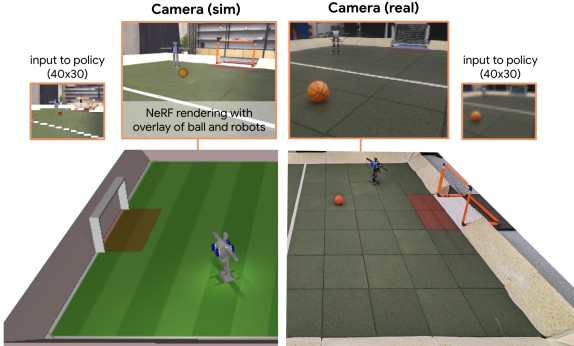

Figure 1: **Environment**. We train agents purely in simulation and align the simulated environment (left) with the real-world environment (right), to enable zero-shot transfer. In simulation, the camera view consists of a NeRF rendering of the static scene (i.e., the soccer pitch and background), with the dynamic objects overlaid. In the real world environment, which is 5m by 4m, we use the output of the head-mounted RGB camera. Agents receive downsampled $40 \times 30$ resolution images.

In this work, we present an RL pipeline for training vision-based policies for multi-agent robot soccer without any of these simplifying assumptions. Our policies use *purely onboard sensing*, consisting of an IMU, joint encoders, and a head-mounted RGB camera. Robot soccer is a challenging domain that requires not only agile low-level control, but also long-horizon planning for positioning, predicting and scoring against an opponent. Many of these challenges have been addressed by previous work [4] using ground-truth state information. This paper builds on that work, but focuses specifically on challenges that arise when perception is restricted to onboard sensors including, in particular, egocentric RGB vision.

Egocentric vision renders the environment partially observed, amplifying challenges of credit assignment and exploration, requiring the use of memory and the discovery of suitable information-seeking strategies in order to self-localize, find the ball, avoid the opponent, and score into the correct goal. A moving opponent and ball add further complexity to the limited field of view and potential motion blur that vision-based policies must contend with. To perform well, agents must learn to combine agility and precise control (for example, to kick a moving ball) together with information-seeking behavior via active vision. This combination of behaviors is difficult to manually script, due to the existence of multiple simultaneous objectives, the dynamic environment, and the highly context-dependent nature of the optimal behavior.

We present a method for training vision-based RL agents end-to-end from pixels and proprioception for one-vs-one soccer using the MuJoCo simulator [8] and Neural Radiance Fields (NeRF) to provide a realistic rendering of the real physical scene [9, 10] (shown in Figure 1) which enables zero-shot transfer to the real world. We combine agent experience across training iterations, to improve learning speed and asymptotic performance [11]. Importantly, our approach does not involve any changes to the task or reward structure, makes no simplifying assumptions for state-estimation, and does not use any domain-specific architectural components.

We observe strong agent performance and agility even in this partially-observable and dynamic setting. Our vision-based agents walk as fast and kick as strongly as a state-based agent that has access to ground-truth egocentric locations of the opponent, ball, and goal. While our training setup includes no explicit rewards for active perception, information-seeking behaviors such as searching for the ball emerge naturally during training, from the simple incentive to play soccer well. Our analysis shows that the agents are able to track moving objects even when occluded and out-of-view in both simulation and the real world, and have a propensity to track moving balls of various shapes and colors. They demonstrate complex opponent-aware behaviors like blocking and shielding the ball and accurately score even when the goal is out of view. Our accompanying website (https://sites.google.com/view/vision-soccer) includes video analysis and full games showing agile soccer play from vision.

The key contributions of this work are as follows:

- A soccer agent trained end-to-end with RL using RGB pixels; a first demonstration of this kind on this challenging partially-observable task.

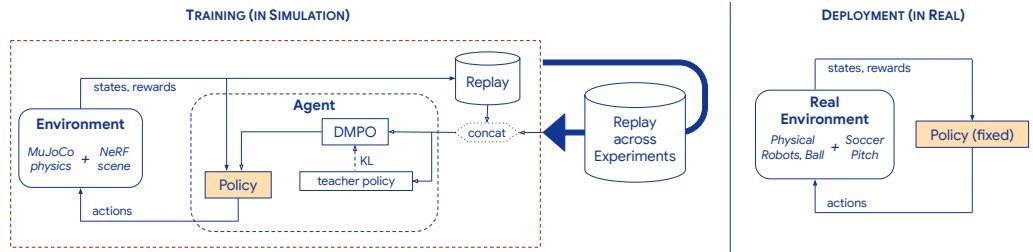

Figure 2: **Method Overview**. In each experiment, we train the agent with off-policy RL via Replay across Experiments (RaE) [11] from a mixture of current experience and previously collected data. We additionally regularize the agent in later training stages towards teacher policies. The agent is trained purely in simulation, and transferred zero-shot to the physical world. The agent only perceives onboard observations, consisting of RGB images and proprioception.

- A detailed analysis of soccer play and agent behavior with a focus on emergent vision-specific skills, including active perception of the ball and opponent tracking.
- A quantitative comparison that shows our vision-based agents maintain a similar level of agility to state-of-the-art agents with access to ground-truth state information.

## 2 Method

In this section we present our approach to learning RGB-vision based soccer. As discussed in Section 1, learning to play soccer from vision is substantially more challenging and requires different information-seeking optimal strategies compared to learning with access to ground-truth state. Therefore, we propose a training pipeline that focuses on learning end-to-end from vision, rather than distilling vision policies using state-based agents [6, 12].[4] Our approach uses the two-stage RL method described in Haarnoja et al. [4] which we summarize in Section 2.1, but with key additional components such as the use of Replay across Experiments (RaE) [11] to increase data usage efficiency and the use of calibrated NeRF rendering [9, 10] with multiple NeRFs which we discuss in Section 2.2. The soccer agents are trained entirely in simulation using MuJoCo [8], and deployed zero-shot on the real platform (Figure 2).

### 2.1 Two-Stage Training with Reinforcement Learning

Our method uses the two-stage reinforcement learning pipeline of Haarnoja et al. [4], which uses the Maximum a-posteriori Policy Optimization (MPO) algorithm [13] and a distributional critic [14]. In the first stage, two separate experts are trained: one to get up from the ground and another to score against a fixed, random opponent. In the second stage, these experts are distilled into one agent using RL with adaptive KL-regularization. In this stage, the opponent is randomly selected from the first quarter of the agent's saved policy snapshots. This ensures that the agent progressively plays against increasingly challenging opponents, and encourages learning robust multi-agent strategies. Random perturbations and physics randomization are used to improve zero-shot transfer to the real world. In contrast to the original pipeline [4], which does not rely on long-term memory, we use an Long Short-Term Memory (LSTM) architecture [15] with a long unroll length to incorporate memory over time and permissive head joint limits to allow more freedom for active perceptual behaviors (see Appendix A).

### 2.2 Training from Egocentric Vision

When training from vision, we provide the critic with ground-truth state information [16] including the egocentric position and velocity of the ball, opponent, and goal and the visual and proprioceptive

---

[4]We compare the benefits of our approach to state-to-vision transfer empirically in Section 3.4.

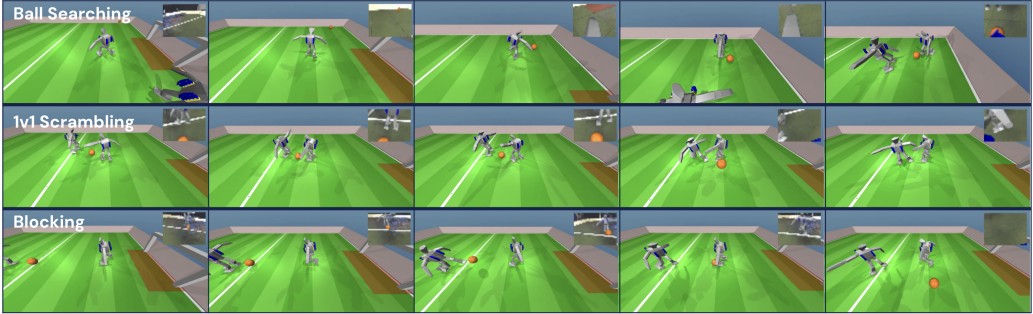

Figure 3: **Emergent behaviors**. Each row shows a different emergent behavior. The agent's camera view is at the top right of each frame. **Top row:** The agent pivots to scan the scene and localize the ball, walks towards it, and positions itself to shoot. **Middle:** Agents scramble for the ball and try to shoot past each other. **Bottom**: The agent positions itself to prevent the opponent from scoring.

information that is provided to the policy. This simplification exploits the fact that the critic is not required at deployment. In addition, we introduce further key components to enable learning from vision which we describe below:

**Rendering**   For realistic camera observations, we rely on real-to-sim transfer by generating various NeRFs of the real soccer scene [9, 10] using 250-300 photographs. This enables rendering the static scene from novel viewpoints. We render dynamic objects such as the ball and opponent robot using MuJoCo and overlay them on the static scene to produce a final composite camera view. During training, a scaled down version (to 40x30 pixels) of this image is provided as visual input to the policy, as shown in Figure 1. To capture the NeRF we use a Sony $\alpha$7C camera which differs from the robot's camera, a Logitech C920 webcam. To reduce this sim-to-real gap, we calibrate the NeRF colors using a linear transformation (see Appendix A).

**NeRF Randomization**   We apply various randomizations to the agent's visual observations including commonly used image augmentations like random perturbations to the brightness, saturation, hue, and contrast of the input. In addition, we collect 4 NeRFs under differing lighting conditions and with different arrangements of background objects, and randomly select from them, per episode, during training. This simple addition introduces variations to the scene which we leverage as an additional form of domain randomization.

**Data Transfer and Offline-Online Mixing**   Training from vision is considerably more complex, expensive and time-consuming than from state: actors must render at each step and are slower to generate data, and each batch of learning data is slower to process. To reduce iteration time and use visually diverse data for training (to reduce visual overfitting), rather than training from scratch on every iteration, we reuse transition data from previous experiments (with different policies, hyperparameter settings etc.) [11]. This addition significantly improved asymptotic performance and enabled learning across both stages of the training pipeline.

## 3   Experiments

We deploy policies on Robotis OP3 humanoid robots [17], on a pitch that is 4 m wide by 5 m long (Figure 1). The robots are 51 cm tall, weigh 3.5 kg, and have 20 actuated joints. We add minor modifications, including 3D-printed bumpers (which are also rendered), to increase physical robustness [4]. The robot's sensors consist of joint position encoders, an IMU, and RGB images from a head-mounted Logitech C920 camera— running at 30 frames per second. Images are downsampled to $40 \times 30$ to reduce onboard policy inference time. The robot's head controls the camera pan and

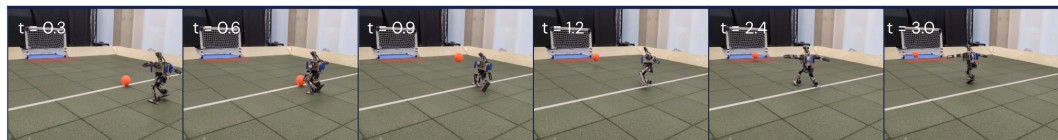

Figure 4: **Shooting behavior, real-world:** The agent smoothly transitions between skills and scores.

tilt. We use 40 Hz joint position control [5], and smooth actions with an exponential filter[6] to avoid damaging the actuators with high-frequency action variation.

Our experimental analyses seeks to answer the following questions:

- What behaviors emerge naturally from training vision-based agents with RL to play soccer?
- Can memory-augmented agents robustly track objects (e.g. the ball and opponent) in scenes with partial observability and occlusion?
- Does training from vision affect agility, compared to training with privileged information?
- Instead of training entirely from vision, can we distill knowledge from state-based agents?

We investigate these questions with comparisons to a "state-based" agent with access to ground-truth state information at test time, consisting of egocentric position and velocity of the ball, opponent, and goal. For complementary videos, please refer to our accompanying website.

## 3.1 Soccer Behaviors

Figure 3 highlights behaviors that emerge while training agents in simulation: searching for the ball, scrambling, and blocking a shot. These complex, long-horizon, multi-agent behaviors rely on active vision and are difficult to manually script. Interestingly, such behaviors emerge naturally from the soccer task reward, and without the need for explicit reward terms for, say, finding the ball.

Arguably the most important skill for a soccer agent is the ability to shoot and score. Figure 4 shows an example of the learned shooting behavior, transferred zero-shot to the real robot. Classical approaches sequentially compose individual behaviors, often with rough transitions between different motions. In contrast, here the agent smoothly transitions between walking towards the ball, positioning itself, and kicking without pauses or discontinuities. Section 3.3 presents a quantitative analysis of this shooting behavior compared to the state-based agent.

## 3.2 Object Tracking from Vision

**Agent Representation Probes** To probe the agent's object tracking ability, we train linear layers downstream from the frozen encoder features of the policy [18], to predict the positions of various objects using Gaussian mixture models (see Appendix A).

Figure 5 visualizes these predictions in simulation using a heatmap, and shows the predicted positions of the agent, opponent, and ball are fairly accurate and become more confident over time. In the first row, the agent localizes its position using distinguishing features of the environment (e.g. the orange goal) and remembers its location even when these features are out of view. Although opponent tracking is less accurate, a similar tracking ability emerges. Ball tracking is remarkably accurate: the agent precisely locates the ball once it is in view and, importantly, is able to predict ball movement (and hence implicitly velocity) even after kicking it out of view. We observe similar results in the single-scene real world analysis of Figure 6. This suggests that object tracking is robust to visual changes (like lighting and camera blur) that arise when using the onboard camera, albeit with a slight decrease in prediction accuracy. Overall these analyses indicate that tracking dynamic objects in the visual scene emerges naturally via our training pipeline.

---

[5]The different rates for policy and camera imply that the visual input may be held constant across two timesteps which we do not find that this present an issue in practice.

[6]$u_t = 0.8u_{t-1} + 0.2a_t$, where $a_t$ is the policy output and $u_t$ is the action applied to the robot at timestep $t$.

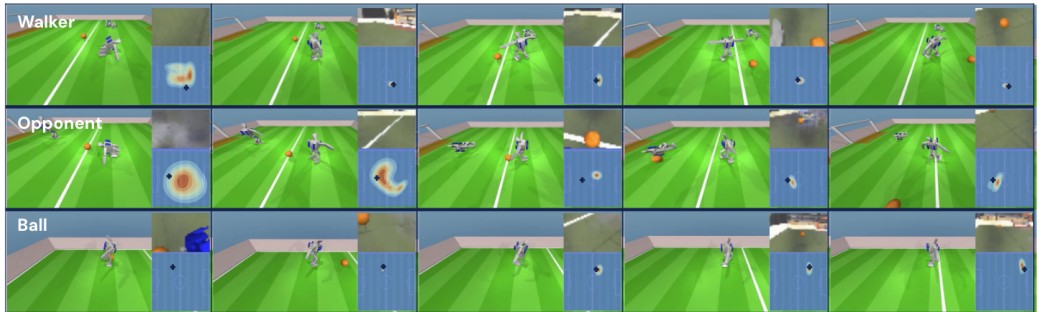

Figure 5: **Position prediction results:** Each row shows the predicted positions of either the agent, opponent, or ball across time. Each frame shows the simulation with the egocentric camera view in the top right and the predicted quantity in the bottom right. The heatmap of the Gaussian mixture model converges to a point as the certainty increases. Ground truth is indicated by a black cross.

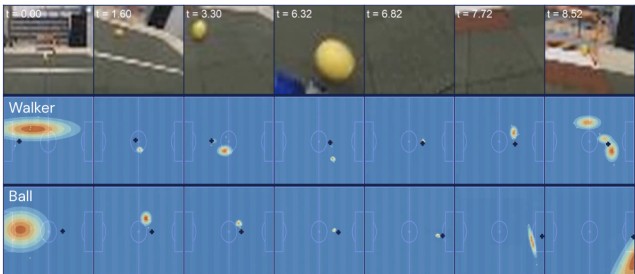

Figure 6: **Position prediction results, real world:** Each column displays the egocentric camera view (top), predicted agent (middle) and ball (bottom) position from the same instant in time. Initially the agent is uncertain about both its own position and the ball's position. After finding the ball the predictions become more certain and are relatively accurate. After kicking the ball (column 4), the agent continues to accurately predict the ball location even when the ball is no longer in view.

**Head Movement Probes** Qualitative observations of the agent's behavior during game play suggest it actively moves its head camera to track the ball. To study this behavior in isolation, we initialize the robot in simulation in a standing position with the ball directly in front of it and moving with a random initial velocity of up to 1 m/s. We only allow the agent to move the head pan and tilt joints, which control the camera orientation. We then measure the angular distance between the center of the camera gaze and the location of the ball across 16 episodes of 100 timesteps each. Compared to a baseline where the head is fixed, the agent moves the camera to ensure the ball remains in the field of view (Figure 7a).

We observed the same head tracking behavior on the real robot, where the agent robustly moved its head to follow moving balls with a range of colors and sizes (Figure 7b). We do not directly incentivize the agent to localize the ball; this example of active perception emerges naturally from training on the soccer task reward.

### 3.3 Analysis of Agility

In this section we evaluate our vision-based agents on agility and scoring ability. We follow the setup described in [4], with minor modifications to ensure a fair comparison (see Appendix B).

In terms of agility, the vision-based agent fares better across all metrics (walking speed, turning speed, and kicking power) when compared to the scripted baseline and is on-par with the state-based agent for walking speed and kicking power (Table 1).[7] This indicates that comparable agility can be achieved from onboard sensing alone.

---

[7]Note that turning speed is measured by initializing the agent and ball on opposite corners of the pitch, with the agent facing away from the ball. A state-based agent knows the location of the ball behind it and thus turns

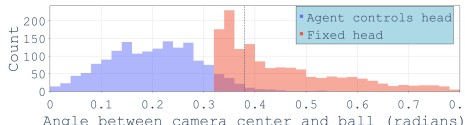

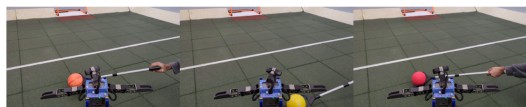

(a) **Active ball tracking**  (b) **Real-world head tracking**

Figure 7: (Left): Histograms of the angular distance between the center of the camera view and the ground-truth ball location, over 16 episodes. The vertical line indicates the approximate size of the camera's field of view. In the red control group, the camera is fixed facing forward while in the blue trajectories the learned policy is allowed to only move the head. (Right): We freeze all joints except head pan and tilt to demonstrate the learned use of controlling the head camera for active perception. Ball color randomization ensures the agent is able to track balls with a range of colors.

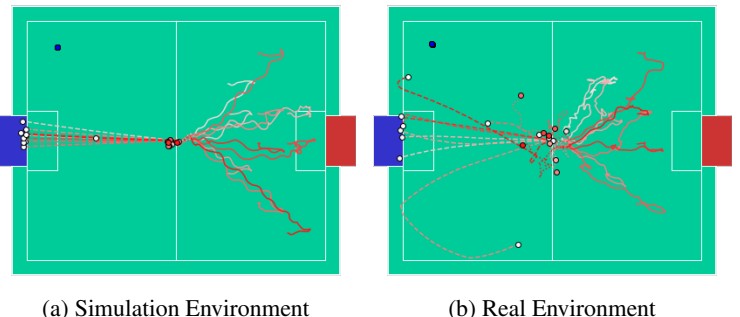

(a) Simulation Environment  (b) Real Environment

Figure 8: **Scoring ability:** Top down view of simulated (left) and real-world (right) penalty shoots. The opponent (blue circle) is stationary, the agent (red circle) is initialized on the floor at random positions in its own half, and the ball (white circle) is initialized in the pitch center. Solid lines show the agent's path and dotted lines show the ball's path. The target goal is blue, and the agent's own goal is red. Positions for the real environment (determined via a motion capture system), are used purely for analysis. Scoring in simulation is near-perfect with minimal variance whereas scoring in real fares worse. In the real world, robot behavior is more stochastic, and both agent and ball paths are affected by the slightly uneven ground.

We measure scoring ability across 250 penalty shoots in simulation and 20 in the real world, with trajectories visualized in Figure 8. The agent is initialized on the ground in a random initial position and is given 12 seconds to score. In simulation, the vision- and state-based agents have similar scoring accuracy, an average of $0.86$ versus $0.82$. In the real world however, both agents suffer from a drop in performance, with a lower average accuracy of $0.4$ for the vision-based agent versus $0.58$ for the state-based agent. For the state-based agent, in Haarnoja et al. [4] this drop in real-world performance was largely attributed to sources of noise that are difficult to simulate, such as sensor noise, battery state, and the state of the robot hardware. This is exacerbated for vision agents, with additional sources of noise including changes in lighting conditions and motion blur.

| Metrics | Scripted | State-based | Vision-based |
|---|---|---|---|
| Walking Speed | $0.20 \pm 0.05$m/s | $0.51 \pm 0.01$m/s | $0.52\pm0.02$m/s |
| Turning Speed | $0.71\pm0.04$rad/s | $3.17\pm 0.12$rad/s | $2.78\pm0.08$rad/s |
| Kicking Power | $2.07 \pm 0.05$m/s | $2.03 \pm 0.19$m/s | $1.95 \pm 0.31$m/s |
| Scoring (sim) | - | $0.82 \pm 0.05$ | $0.86 \pm 0.04$ |
| Scoring (real) | - | $0.58 \pm 0.07$ | $0.4 \pm 0.11$ |

Table 1: **Analysis of agility:** Comparison of the vision-based agent against state and scripted baselines for behaviors in simulation (top four) and real (bottom). Mean and standard error are across 10 trials for all except *scoring*, which is over 250 episodes in simulation and 20 episodes in real.

---

quickly to arrive there, whereas a vision-based agent is at a disadvantage, since it does not "know" a-priori that it must do a full 180-degree turn.

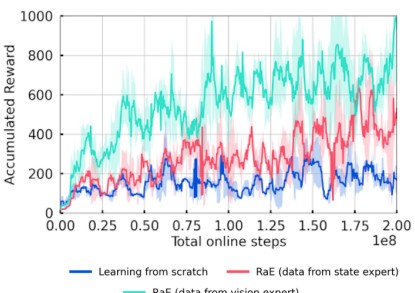

Figure 9: **Data source analysis:** Comparing transfer performance when using data generated by a state-based expert (that also renders camera observations) and data generated by a vision-based agent to learning from scratch.

## 3.4 Data Sources Analysis

Finally in this section we will provide some evidence that argues in favor of training end-to-end from vision over distilling knowledge from state-based agents. For this analysis, we generate gameplay data from a state-based agent while rendering from the agent's camera; which ensures the data is compatible for learning with RaE. We compare this to data generated using our vision learning pipeline. We find that while reusing data generated from a state agent can improve performance, there is a notable difference to using vision data directly (Figure 9). Intuitively the behaviors discussed in the previous sections require actively controlling and coordinating movements between the head and body: since state-based agents do not require such behaviors, reusing their data may be of limited use.

## 4 Related Work

While several works have developed techniques for sim2real transfer of RL controllers, none have addressed RGB perception with end-to-end RL in a dynamic, long-horizon, multi-agent task like soccer. External state estimation [4, 6] or depth sensing [19, 1, 3] are often used to simplify exteroception while domain randomization [20, 21, 22] has been studied for RGB perception on simpler tasks. Recently, real2sim techniques [23, 24] which can produce realistic visuals in simulation have shown promising results in fully observed tasks like goal-directed navigation [10]. [25] addressed the task of vision-based dribbling with a quadrupedal robot and specific architectural assumptions (e.g. fall detectors to switch to a recovery policy). Our approach is most closely related to the work of Haarnoja et al. [4], Byravan et al. [10] but we consider a more challenging partially observable setting which requires additional domain randomization and training protocols (e.g. data reuse [11]). Consequently, the focus of our analysis are active-vision behaviors which arise without any explicit rewards unlike in prior work [26, 27].

While multi-agent soccer has been studied extensively in simulation and the real world through, in particular, the RoboCup competition [28, 29], our work focuses on a simplified setting (e.g. no fouling, substitutions, set-pieces, coach or communication), with simplified game dynamics (e.g. ramps to rebound balls which are out-of-bounds), to facilitate a more directed study of RL. While RL has been used in specific soccer settings within RoboCup [30, 31, 32, 33, 34, 35], to the best of our knowledge, our work is the first demonstration of real-world transfer for vision-based policies trained with end-to-end RL for the dynamic, long-horizon task of robot soccer. An extended discussion of related work is in Appendix D.

## 5 Conclusions

We introduce a system for vision-based multi-agent soccer via end-to-end RL and zero-shot transfer to real robots. Our vision-guided agents show similar levels of agility as state-based agents and show active-perceptual behaviors with no explicit rewards and use multiple NeRFs and RaE as general tools that enable learning. We provide a list of limitations under Appendix E.

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

# Appendix

## A  Training Details

Our training setup is similar to the one proposed by [4] with a couple of important changes that we detail in this section. Since our training pipeline relies entirely on onboard sensing with the egocentric camera, we do not require motion capture markers for tracking the agent, opponent and ball position. We only use these markers occasionally for analysis; for example to measure the number of goals scored in the penalty set piece. Please find additional videos and descriptions on our website [8].

**Reinforcement Learning setup**  We consider reinforcement learning in a Partially-Observable Markov Decision Process (POMDP) consisting of state space $\mathcal{S}$, action space $\mathcal{A}$, transition probabilities $p(s_{t+1}|s_t, a_t)$, and rewards $r(s_t, a_t)$. The agent perceives observations $o_t$ which only contain part of the state information $s_t$. We consider policies $\pi(a_t|x_t)$ that are history-conditional distributions over a length-$l$ history of previous observations $x_t = (o_{t-l}, ..., o_{t-1}, o_t)$. The goal of RL is to learn an optimal policy $\pi^*$ that maximizes the expected sum of discounted rewards: $\pi^* = \arg\max_\pi (\mathbb{E}_\pi[\sum_t \gamma^t r(s_t, a_t)])$.

**Joint Limits**  We limit the joint range in software as done in [4] but vary the limits of the head joints to allow the walker to more freely search for the ball. The full list of joint limits used in our work is listed in Table 2.

**Reward Description**  We do not modify the reward specification from [4] in any way. We observe information-seeking and visual tracking behaviors emerge naturally through the course of training without the need to specify any additional rewards to encourage this. We refer the reader to [4] for a full description of the rewards used for training.

**NeRF calibration**  To capture the NeRF we use a Sony $\alpha$7C camera which differs from the robot's camera, a Logitech C920 webcam. To reduce this sim-to-real gap, we place the Logitech C920 camera to the center of the soccer pitch and rotate it by 360 degrees while recording a 30fps video (about 30 seconds). Then we record an analogous video inside the NeRF simulation. We calibrate the NeRF colors using a per-pixel and per-channel linear transformation which is learned by matching the per-pixel mean and variance of the RGB colors in the two videos. Specifically, let $\mu_{real}$ and $\mu_{nerf}$ be the average pixel RGB values across all frames in the real and NeRF video respectively, and let $\sigma_{real}$ and $\sigma_{nerf}$ be the standard deviations of the pixel RGB values. When we render from the NeRF, we apply the following transformation to the color of every pixel in every frame:

$$\text{color} = \text{clip}\left[\frac{\sigma_{real}}{\sigma_{nerf}}(\text{color}_{nerf} - \mu_{nerf}) + \mu_{real}, \, 0, \, 255\right]. \tag{1}$$

**Agent Training**  We use the Maximum a-posteriori Policy Optimisation (MPO) algorithm [13] with a categorical distributional critic [14] for training. We condition the critic on privileged information including the egocentric position and 2-D velocities of the opponent and ball and the egocentric locations of the goal. The actor is conditioned on proprioceptive information including joint positions and information from the IMU. Finally both actor and critic are conditioned on a $40 \times 30$ rendering of the egocentric view. During training we sample an image of this resolution from one out of 4 randomly chosen NeRFs of the same scene that were created with variations in background and lighting. The opponent and ball are rendered using Mujoco and overlaid on top of the image generated by the NeRF to obtain the final image for training. During evaluation we downsample images rendered from the onboard camera to a resolution of $40 \times 30$ to run the policy

---

[8]https://sites.google.com/view/vision-soccer

at a 40Hz frequency. Note that the camera operates at a lower frequency of 30Hz which means the visual input to the policy repeats across two consecutive timesteps.

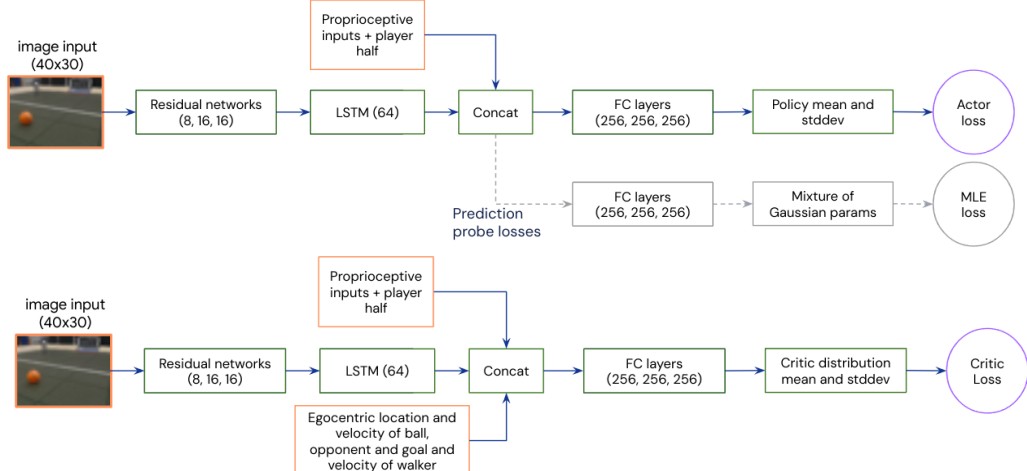

Figure 10: **Network architectures used for training.** The top row shows the architecture used for the policy with the prediction probes discussed in Section 3.2. The bottom row shows the critic architecture.

**Network Architecture** The network architecture we use for training is based on the architecture used in [10] and are shown in Figure 10. As the figure shows, image inputs are encoded using a sequence of 3 residual blocks of size 8, 16 and 16. The output features from this transformation are then processed by an LSTM block of size 64. The output of this is then transformed using an MLP layer to parameterize a categorical distribution for the critic, or a Gaussian distribution in case of the actor. The policy network also shows the prediction heads that are used to predict various quantities like the egocentric position of the ball and goal in Section 3.2. When training the prediction probes the losses only flow through the arrows indicated with a dotted line.

**Distributed Training Setup** We use a distributed training setup with 64 V100 GPU actors and a TPU v2 learner pod arranged in a $2 \times 2$ topology. The data produced by the actors is written to a replay buffer ensuring that on average 16 environment steps are taken for each learner update with the ability to combine these with offline stored datasets as described in the main paper. A full set of hyperparameters used for training are presented in Table 3.

**Ball Randomization** To enable real world application with differently colored balls and to increase robustness to different lighting conditions we apply variations to the ball during training. In particular, we take the base RGB value, radius and weight of the ball in simulation (that were generated using a ground-truth orange ball in the real world) and uniformly sample from distributions ranging from 0.8 to 1.2 times the actual value.

**Training Workflow** As described in the main text, we train expert policies for getting up from the ground and scoring and required one iteration of RaE to generate a successful scoring policy. We then train the final policy using regularization to the experts and reusing the data generated for training the scoring expert. The scoring expert is trained using a random opponent with a an environment termination on falling. We train the final policy for 240,000 episodes with RaE using two datasets of 400,000 episodes and 220,000 episodes (that were generated from training the scoring expert).

The scoring expert policy is trained using a trajectory length of 48 (which is effectively the context window of the LSTM) but to enable longer histories for the final policy, we used a trajectory length of 145. We also use the multi-agent curriculum training strategy as described in the main text to generate the complex behaviors described.

| ID | Joint Name | Min | Max |
|----|-----------|------|-------|
| 19 | head_pan | -2.5 | 2.5 |
| 20 | head_tilt | -1.26 | -0.16 |
| 16 | l_ank_pitch | -0.4 | 1.8 |
| 18 | l_ank_roll | -0.4 | 0.4 |
| 6 | l_el | -1.4 | 0.2 |
| 12 | l_hip_pitch | -1.6 | 0.5 |
| 10 | l_hip_roll | -0.4 | -0.1 |
| 8 | l_hip_yaw | -0.3 | 0.3 |
| 14 | l_knee | -0.2 | 2.2 |
| 2 | l_sho_pitch | -2.2 | 2.2 |
| 4 | l_sho_roll | -0.8 | 1.6 |
| 16 | r_ank_pitch | -1.8 | 0.4 |
| 17 | r_ank_roll | -0.4 | 0.4 |
| 5 | r_el | -0.2 | 1.4 |
| 11 | r_hip_pitch | -0.5 | 1.6 |
| 9 | r_hip_roll | 0.1 | 0.4 |
| 7 | r_hip_yaw | -0.3 | 0.3 |
| 13 | r_knee | -2.2 | 0.2 |
| 1 | r_sho_pitch | -2.2 | 2.2 |
| 3 | r_sho_roll | -1.6 | 0.8 |

Table 2: Joint Limits

| Parameter | Value(s) |
|-----------|----------|
| Batch size | 80 |
| Trajectory length | 48, 145 |
| Max replay size | 100000 |
| Actor learning rate (ADAM) | 1e-4 |
| Critic learning rate (ADAM) | 1e-4 |
| Temperature learning rate (ADAM) | 1e-2 |
| Tradeoff learning rate (MPO) | 1e-4 |
| Action samples (MPO) | 20 |
| $\epsilon$ KL Mean (MPO) | 0.0025 |
| $\epsilon$ KL Covariance (MPO) | 1e-6 |
| Discount factor | 0.99 |

Table 3: Hyperparameters used during training

# B  Experiment Details

We perform set piece tests coarsely in line with prior work on evaluating state based soccer agents. However, based on camera resolution provided to the agent we need to adjust the Walking Speed set-piece. Instead of placing the agent at position (0.1, 0.5) relative to the pitch and ball at the end of the pitch at a location of (0.9, 0.5), we place the ball at position (0.7, 0.5) since the original location did not enable the agent to see, and therefore walk to, the ball.

# C  Additional Experiments

## C.1  Privileged Information for Asymmetric Actor Critic

Figure 11 compares learning performance with and without passing additional privileged ground-truth state information of the location of the opponent and ball. We find that learning is significantly affected by this choice and performance without the ground truth information remains very low.

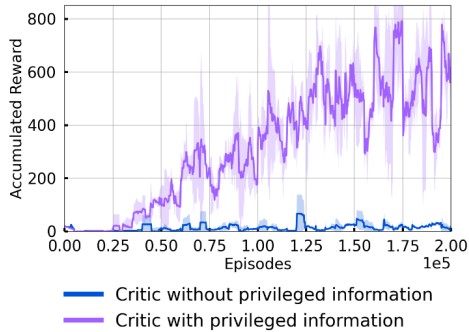

Figure 11: Critic with privileged information including ground truth states and without. Performance without privileged information is considerably lower.

## C.2    Scoring Comparison Between State and Vision-based Soccer

A direct comparison between state- versus vision-based agents is heavily biased against vision-based agents: state-based agents are given ground-truth locations of the opponent, ball, and goal, whereas vision-based agents must infer this from vision and proprioceptive observations. Our analysis in the main text shows that despite solving this more challenging task, vision-based agents maintain similar levels of agility as state-based ones.

However, when we match the agents head-to-head, the state-based agent's knowledge of ground-truth information enables it to more quickly reach the ball and attempt to score. We ran an analysis comparing vision- and state-based agents over 100 matches in simulation, where each match lasts up to 40 seconds or terminates early when an agent scores. We found that averaged across the games the vision-based agent won 27 percent of the games, indicating that the state-based agent wins about 3 times as often which serves as a reference for the scale of the challenge faced when playing from visual inputs.

## D    Extended Related Work

**Sim2Real Transfer for Robot Locomotion**  Several works have developed techniques for sim2real transfer of locomotion controllers, e.g. [36, 37, 38, 19, 39, 4, 1, 3]. However, few works have addressed the complexity of transfer with RGB perception when applying end-to-end RL in a dynamic, long-horizon task like soccer. Some approaches partly address this challenge by replacing RGB perception with external state estimation [4], height sensors [19], or depth sensors [1, 3]. On the other hand, other approaches have addressed the problem of transfer with RGB-based perception but in less challenging domains. Early such approaches studied domain randomization, where randomizing lighting, textures and other rendering options [20] can improve transfer (see [21, 22] for a review). Rendering based randomization usually involves task-specific, engineering heavy processes which are typically more involved and require variation in the shapes, position and textures of objects (and distractors) in the scene as well as characteristics of the camera like lighting, location and orientation [20, 6]. More recently real2sim techniques which rely on a small amount of real data to accurately reproduce scene visuals in simulation have become popular [23, 24, 10] and have shown promising results in tasks like goal-directed navigation and dribbling but do not address challenges like partial observability and long task horizons that are present in soccer. Our approach is inspired by and builds upon these ideas be relying on a number of NeRFs to recreate visual characteristics that the agent is likely to encounter in the real world. Finally, [25] introduced a pipeline for training a quadruped robot that is capable of dribbling a soccer ball from visual inputs. Their pipeline demonstrates remarkably robust behavior but is less general and relies on specific architectural assumptions (e.g., a fall detector to switch to a recovery policy).

Our work builds on and extends the domain randomization and system identification protocols of Haarnoja et al. [4], Byravan et al. [10]. While our approach is inspired by and combines previous work, to the best of our knowledge, our work is the first demonstration of real-world transfer for vision-based policies trained with end-to-end RL for a dynamic, long-horizon task.

**Multi-Agent Reinforcement Learning**   Competitive play has historically been used to demonstrate breakthroughs in artificial intelligence [40, 41, 42, 43, 44] and, more recently, as a mechanism that leads to emergent capabilities [45, 46, 47]. This research continues that line of work beyond simulation and simple embodiments to real robots in a partially-observed environment with egocentric vision. Learning in non-stationary competitive environments can be unstable [e.g. 48], and several works [e.g. 49, 50] have applied techniques from game theory [51] to reinforcement learning to stabilize training and reach principled equilibria as solutions. Similarly, [44] trained against a league of opponents. Our more practical approach, motivated by those ideas, also aims to achieve robustness against a variety of opponents.

**Skill and Data Transfer**   Knowledge transfer in RL agents has been facilitated in various ways including the reuse of data or using agent policies or skills to accelerate learning and asymptotically improve performance [52, 53, 54]. The idea of composing together previously learnt skills has been explored using regularization [55, 56, 57] or in the context of Hierarchical RL where a high level controller leans to compose behaviors together to learn new solutions [58, 59, 60, 61, 62, 63]. Knowledge transfer can also be enabled through the use of previously collected data to bootstrap learning; either as expert data to bootstrap learning [64, 65, 66, 67, 68] or by learning from offline datasets [66, 69, 70, 71, 72]. In this work, we leverage both forms of knowledge transfer by regularizing to previous teachers [73, 4] and using previously collected data to bootstrap learning [11].

**Robot Learning and Learning in Robocup**   Isolated skills such as running, climbing and jumping have been demonstrated on bipedal robots [74, 75, 76, 77, 78, 79, 37, 39]. Our work tackles the problem of robot soccer requiring integration of multiple skills, and uses egocentric vision. The RoboCup competition [28, 29] uses simulated and real soccer environments in a multi-player soccer game with comprehensive rules. In comparison, our work focuses on 1v1 play under simplified rules (e.g. no fouling, substitutions, set-pieces, coach or communication) and with simplified game dynamics (e.g. ramps to rebound balls which are out-of-bounds) in order to facilitate a more straightforward study of fully learned approaches. Reinforcement learning based approaches have been successful in the 2D [30] and 3D simulation leagues [31, 32], where skills learned with RL are combined using a hierarchical system. Reinforcement learning has also been used to train specific isolated soccer skills on real quadruped [33, 34] and wheeled [35] embodiments. In contrast, our work presents an end-to-end system in which perception, low-level motor skills and long-term planning are integrated, and with a real humanoid embodiment. Similar work was presented in [4], but we consider the more challenging setting of using egocentric vision rather than external state estimation. Importantly, our solution relies on general algorithmic improvements (reuse of data and skills along with improved visual domain randomization) and makes no modifications to the task or reward structure.

Many works have used RL to learn active vision policies, e.g. recently [26] for general manipulation and [27] for viewpoint selection in the specific context of RoboCup. These works address the specific challenge of controlling the agent camera or choosing a viewpoint by using particular information-seeking reward functions. By contrast, in this work, active perception behaviors emerge naturally with no direct reward term for perception and are thus smoothly integrated in the full task of soccer.

# E   Limitations

While our work shows promising results for 1v1 soccer from egocentric vision, there are a number of limitations in the current system that can be considered in future work. For instance, the agent's visual policies are conditioned on low-resolution images ($40 \times 30$) which allow for faster training and inference at deployment but may lose important information present in the environment particularly for more complex tasks. In addition, changes in the environment scene that are not modeled by the set of NeRFs will reduce the reliability of the policy. These considerations may become more important when considering a full multi-agent soccer game where it is important to be able to differentiate between opponent agents and team-mates and infer strategies to inform game-play. In addition, the current limited horizon for training LSTM-based agents may also be insufficient to learn about complex, long-term interactions.

