# OpenReview forum: "Learning Robot Soccer from Egocentric Vision with Deep Reinforcement Learning"
_robot-learning.org/CoRL/2024/Conference — CoRL 2024_

### Official Review · Reviewer_PVvE · 2024-07-04
**This paper demonstrates a major advancement on an end-to-end active-vision deep RL problem with the real hardware. Qualitative analysis is well provided in the paper. However, some critical quantitative experiments are missing.**

**Originality:** 3
**Technical Quality:** 3
**Clarity Of Presentation:** 4
**Potential Impact:** 4
**Recommendation:** 4
**Confidence:** 4

**Review:**

The paper is well written and details of the proposed approach is clearly described. Authors tackled an egocentric active-vision task with the real robot, which is considered to be one of the most challenging problems in RL domain. The paper clealy shows that the trained agent demonstrates its capability and is comparable to the previous state-based agent. However, as I described below, some important ablation studies are completely missing in the paper.

Strength:
* The approach is a natural extension to the previous state-based agent in the same domain.
* The resulted system is simple and capable of executing the challenging egocentric active-vision robotic task.
* The use of RaE in the two stage training is clever since it is not introducing additional complexites in the training algorithm.
* Thorough qualitative analysis is provided to show the active-vision ability of the agent.
* The proposed method is evaluated quantitatively to show that the vision-based agent is comparable against the state-based agent.

Weakness:
* The paper is lacking experimental results to show how individual choices contribute to the final performance. For example, how much does the privileged information in critic help? Most importantly, how much does NeRF contribute to the performance? Is simple domain randomization on rendering parameters not sufficient? This analysis is completely missing even though the use of NeRF is core component of the proposed training methodology.

**Quality Of The Limitations Section:**

3

**Questions For Rebuttal:**

I'd like to see analysis of NeRF components in the proposed training methodology. More specifically, I'd like authors to show that the NeRF component is essential to achieve the competitive performance. There is a chance that the NeRF component could be unnecesssary because resolution of the input image is very low, which would make it possible that simple domain randomization could lead to the same level of performance.

Other than that, I have minor questions:
- Accoring to the description, the webcam is running at 30Hz while the agent is acting at 40Hz. This means that the agent sees the latest captured image, but the image could be the same as the previous step? This sort of details will be helpful for practioners who try to apply deep RL to robots.
- `Max replay size` is 100,000, which is smaller than the number of training steps. Is there any reason for this choice?

**Robotics Focus:**

4

**Summary Of Paper:**

This paper proposes a method to train an agent capable of playing soccer with the real robot, using two-stage RL training with a replay across experiment scheme (RaE) and a NeRF-augumented simulation environment. Notably, this paper demonstrates the first example that solves the problem with a challenging egocentric active-vision setup.

**Summary Of Recommendation:**

The demonstration in this paper is considered to be a major milestone of advancements of deep RL with the real hardware. And, the end-to-end active-vision problem will be an important deep RL topic for the next few years. However, the paper is missing analysis about its core component, which could be critical for the decision.

---

### Official Review · Reviewer_4VaL · 2024-07-20

**Originality:** 4
**Technical Quality:** 5
**Clarity Of Presentation:** 5
**Potential Impact:** 3
**Recommendation:** 3
**Confidence:** 2

**Review:**

The paper is closely related to the work of Haarnoja et al. [4], using the same setting. However, it introduces a significantly more complex task - using egocentric RGB vision instead of ground-truth state information and proposes a successful strategy to solve it - with the help of NeRFs randomizations. The proposed methods are convincing, and results are well supported with hardware demonstrations.

The paper has several good ablation studies. It demonstrates the emergence of complex strategies which, in my opinion, support the point of training agile and competitive agents. It is nice that authors dig deeper into the agent's internal world perception showing how it can predict the position of important objects and follow the ball with its head.

I agree that speed, kicking power and scoring are good quantities to measure the agility of the agent, however, I would like to see some qualitative comparison between State-based and Vision-based agents. How do changes in these metrics influence the win-probability of the agent?

**Quality Of The Limitations Section:**

3

**Questions For Rebuttal:**

* I would like to see the comparison of Visual-based and State-based agents in terms of win-probability. To better understand how it relates to speed and scoring metrics that you have stated.
* Typo on line 548: “lSTM” instead of LSTM

**Robotics Focus:**

4

**Summary Of Paper:**

This work presents an approach of end-to-end robot soccer agent training from visual inputs. The paper uses randomised NeRFs of the real soccer scene in pairs with dynamic objects acquired with MuJuCo to produce camera views in simulation. The resulting Vision-based agent is able to exhibit complex behavioural patterns like blocking and ball searching and predict positions of important dynamic objects like ball and opponent. The paper compares the agent with State-based one and shows that they are close in terms of speed and kick power however differ in scoring.

**Summary Of Recommendation:**

The paper is convincing and well written, it successfully trains RL agents with RGB vision. Results are extensively demonstrated on a hardware.

---

### Official Review · Reviewer_arj7 · 2024-07-21
**The paper shows effective multi-agent RL training for robot soccer with onboard sensing and NeRF-based simulation for real-world deployment.**

**Originality:** 3
**Technical Quality:** 5
**Clarity Of Presentation:** 4
**Potential Impact:** 4
**Recommendation:** 4
**Confidence:** 4

**Review:**

Strengths:
- The paper successfully demonstrates the use of multi-agent RL for training robot soccer policies using only onboard sensing, which is a complex and dynamic real-world task.
- The combination of NeRF-based simulation and real-to-sim transfer techniques provides a robust framework for zero-shot deployment in real-world scenarios.
- The experiments are comprehensive, showcasing the agent's ability to perform agile maneuvers, object tracking, and strategic behaviors like shooting and blocking without explicit rewards for these actions.
- The approach avoids simplifying assumptions commonly used in other works, such as external state estimation or modular architectures, enhancing the general applicability of the findings.
- The analysis includes a detailed examination of emergent behaviors and a quantitative comparison of performance metrics, providing strong evidence of the efficacy of the proposed method.

Weaknesses:
- The reliance on low-resolution visual inputs (40x30 pixels) might limit the agent's performance in more complex or larger-scale environments.

**Quality Of The Limitations Section:**

3

**Questions For Rebuttal:**

In Figure 10, the residual networks (8, 16, 16) and LSTM (64) networks are used for both the actor and the critic for image information extraction. Are these networks sharing weights? If not, it would be beneficial to justify the design choice.

**Robotics Focus:**

4

**Summary Of Paper:**

This paper presents a study on learning robot soccer using egocentric vision with deep reinforcement learning (RL). The approach uses multi-agent RL to train end-to-end soccer policies that rely on onboard computation and sensing through an RGB camera. The training, conducted in a simulated environment enhanced by Neural Radiance Fields (NeRFs) for visual realism, aims to achieve sophisticated soccer strategies transferable to real robots with low-cost sensors. The paper demonstrates that the trained agents can perform at levels comparable to those with access to privileged information, marking a significant achievement in end-to-end training for multi-agent robot soccer.

**Summary Of Recommendation:**

The paper presents a significant advancement in using multi-agent reinforcement learning (RL) for training robot soccer policies with onboard sensing, leveraging NeRF-based scene representation for real-world deployment. The experiments are thorough, demonstrating the agent's capabilities in performing agile maneuvers and strategic behaviors.

---

### Author Rebuttal · Authors · 2024-08-08

Thanks to all the reviewers for the constructive feedback. We have updated the paper (attached) based on the collective feedback, with changes highlighted in blue, and address reviewer-specific comments separately below. Apart from clarifications and corrections, we have added new experiments (Appendix C), a discussion of the importance of training the critic with privileged information, quantitative and qualitative results from a 1-versus-1 match between a state-based and vision-based agent in simulation, and a discussion of the benefits of NeRF-based training. We thank the reviewers again for the valuable feedback and are happy to discuss further during the rebuttal period.

---

### Decision · Program_Chairs · 2024-09-04

**Decision:**

Accept

**Comment:**

The paper describes a method for using multi-agent RL to play robot soccer directly from pixels. The authors describe a detailed sim-to-real pipeline, with a lot of interesting and important ideas. The task is challenging, the results are impressive, and it seems likely to be a significant work in multi-agent RL for robotics.

The reviews are largely in agreement on the quality of this work. Follow-ups from the authors clarified a few interesting details and helped solidify this impression.

Strengths:
- Impressive results showing multi-robot RL for playing soccer using only onboard sensing
- Robust sim-to-real pipeline
- Reviews agree approach is both straightforward and interesting
- Good evaluations and ablations

Weaknesses:
- No significant weaknesses; see the reviewers' comments for some minor points worth discussing